# Opportunistic CT for Prediction of Adverse Postoperative Events in Patients with Spinal Metastases

**DOI:** 10.3390/diagnostics14080844

**Published:** 2024-04-19

**Authors:** Neal D. Kapoor, Olivier Q. Groot, Colleen G. Buckless, Peter K. Twining, Michiel E. R. Bongers, Stein J. Janssen, Joseph H. Schwab, Martin Torriani, Miriam A. Bredella

**Affiliations:** 1Department of Orthopaedics, Cleveland Clinic Akron General, Akron, OH 44307, USA; 2Department of Orthopaedic Surgery—Orthopaedic Oncology Service, Massachusetts General Hospital—Harvard Medical School, Boston, MA 02114, USA; 3Division of Musculoskeletal Imaging and Intervention, Department of Radiology, Massachusetts General Hospital—Harvard Medical School, Boston, MA 02115, USAmiriam.bredella@nyulangone.org (M.A.B.); 4Department of Orthopaedic Surgery, Amsterdam Movement Sciences, Amsterdam University Medical Center, University of Amsterdam, 1012 WP Amsterdam, The Netherlands; 5Department of Radiology, NYU Grossman School of Medicine, New York, NY 10016, USA

**Keywords:** computed tomography (CT), body composition, opportunistic imaging, adverse events, outcome, spine, surgery, length of stay, complications, reoperation, metastases

## Abstract

The purpose of this study was to assess the value of body composition measures obtained from opportunistic abdominal computed tomography (CT) in order to predict hospital length of stay (LOS), 30-day postoperative complications, and reoperations in patients undergoing surgery for spinal metastases. 196 patients underwent CT of the abdomen within three months of surgery for spinal metastases. Automated body composition segmentation and quantifications of the cross-sectional areas (CSA) of abdominal visceral and subcutaneous adipose tissue and abdominal skeletal muscle was performed. From this, 31% (61) of patients had postoperative complications within 30 days, and 16% (31) of patients underwent reoperation. Lower muscle CSA was associated with increased postoperative complications within 30 days (OR [95% CI] = 0.99 [0.98–0.99], *p* = 0.03). Through multivariate analysis, it was found that lower muscle CSA was also associated with an increased postoperative complication rate after controlling for the albumin, ASIA score, previous systemic therapy, and thoracic metastases (OR [95% CI] = 0.99 [0.98–0.99], *p* = 0.047). LOS and reoperations were not associated with any body composition measures. Low muscle mass may serve as a biomarker for the prediction of complications in patients with spinal metastases. The routine assessment of muscle mass on opportunistic CTs may help to predict outcomes in these patients.

## 1. Introduction

Medical treatment for patients with cancer has improved considerably over time. As a result, the life expectancy is increasing, and this has the unintended effect of a rising incidence of metastatic disease [1]. In patients with metastatic disease, the spine is a common location, affected in nearly 30% of cases [2,3]. Spinal metastases can have devastating symptoms, including severe pain, paralysis, incontinence, and sexual dysfunction. Surgical intervention is often indicated for either spinal cord compression or spinal instability [1,4,5,6]. Considering the expanding treatment regimens for these complex patients, multidisciplinary teams and patients must together weigh the likelihood of improved outcomes, including preservation or an improved quality of life, against potential postoperative morbidity and complications when contemplating surgical management [7,8]. Multiple tools exist to predict survival in spine metastases (e.g., SORG, NEMS, Bollen prognosis) patients, but none for adverse events [9,10,11,12,13,14,15]. While predicting survival is imperative, other postoperative outcomes, such as the length of stay, complications, and reoperations, are important to consider in order to best serve the patient and maintain their quality of life [14,16,17]. Clinical variables have been explored for risk factors for adverse events, but these lack strong predictive power, and thus no prediction tools have been developed for adverse events [12,18,19,20,21,22,23].

Several studies have demonstrated the use of CTs for routine clinical purposes to assess body composition (so called “opportunistic CT”) in patients with malignancies to predict survival [10,19,20,21,24,25,26]. CT is routinely used in cancer staging and surveillance, and these CTs can be used to assess body composition without additional costs or radiation exposure. This puts body composition measures assessed using opportunistic CTs in a unique category of both being readily available and potentially useful for the prediction of outcomes. Recent studies have also shown body composition assessed using CT to be associated with an increased hospital length of stay (LOS), postoperative complications, readmission, and other adverse outcomes in patients with sarcomas [27,28,29,30,31,32]. However, the predictive value of body composition measures for adverse events in patients with spinal metastases undergoing surgery is unknown. Identifying new predictors for adverse events is needed as there is a paucity of data about predicting adverse events in this patient population.

Machine learning, an application of artificial intelligence technology, is a subset of statistics and computer science that allows for the processing of large amounts of data, and can be used to quickly and accurately define and quantify data from CT scans [16,26,33]. Similar technology has been used to predict other conditions such as hypertension in various populations [34]. This study expands on the growing body of literature investigating the use of machine learning technologies and opportunistic CTs in conjunction with clinical factors to predict outcomes in surgical, orthopedic, and oncologic populations.

The primary aim of this study was to evaluate the value of body composition measurements using opportunistic CTs in patients with spinal metastases undergoing surgery. The primary outcomes were LOS, postoperative complications within 30 days of surgery, and reoperations.

## 2. Materials and Methods

### 2.1. Study Design and Data Sources

This work received approval from the institutional review board, and informed consent was not required for this observational retrospective investigation. Between 1 January 2001 and 31 December 2016, data for this study were collected at a tertiary referral hospital in the United States of America. We adhered to the Strengthening Reporting of Observational Studies in Epidemiology (STROBE) guidelines [35]. This study was funded in part by National Institutes of Health Grant K24DK109940, P30DK040561-25, and UL 1TR002541. The authors declare no conflicts of interest.

### 2.2. Participants and Clinical Characteristics

All patients aged 18 years or older, who underwent surgery for cervical, thoracic, lumbar metastases (including lymphoma and multiple myeloma), or spinal metastases, and had an abdominal CT scan within 3 months prior to surgery were eligible for inclusion in the study. Exclusion criteria comprised (1) revision procedures, (2) vertebroplasty or kyphoplasty, (3) absence of L4 level coverage on the CT scan, and (4) metal artifacts on CT scans hindering analysis. Two authors (NDK, PKT) individually assessed the 364 patients that initially matched the inclusion criteria. Furthermore, 165 of these patients were excluded for not including L4 on the CT scan or not having a CT scan within 3 months. Three more patients were excluded due to artifacts on the scan. For patients with multiple abdominal CT scans, the scan closest to the date of operation was considered. Likewise, only the initial surgery was taken into account for patients with multiple spine surgeries, ensuring adherence to the statistical principle of independence. The study encompassed 196 patients with suitable CT scans for assessing abdominal fat and muscle cross sectional areas (CSA). The clinical characteristics of the study cohort were previously documented [33]; however, data regarding adverse postoperative events were not reported. Treatment decisions were made collaboratively between the patient and surgeon. The surgical approach was determined by the surgeon, considering factors such as the patient’s estimated survival, neurological deficits, level of pain, and spinal stability. Throughout the study period, postoperative care and rehabilitation were customized according to the severity of the disease.

Clinical factors known to be associated with postoperative adverse events were included as explanatory variables via a manual chart review [18,19,22,36,37,38]: age; sex; body mass index (BMI; kg/m^2^); Modified Charlson Comorbidity Index [36]; primary tumor type categorized by Katagiri et al. as slow, moderate, or rapid growth [37]; tumor location; additional metastases; American Spinal Injury Association (ASIA) impairment scale at the time of surgery [39]; previous systemic therapy; previous local radiotherapy; spinal pain; region of spinal metastases (thoracic, lumbar, cervical, or combined); number of spine levels undergoing surgery; type of surgical treatment; surgical approach (posterior, anterior, or combined); the duration of primary tumor diagnosis until metastatic operation (days); completed pathological fracture; and the preoperative albumin level (g/dL) closest to the surgery, with a maximum range of being within two weeks of the operation. The patient’s comorbidity status underwent evaluation utilizing the Modified Charlson Comorbidity Index—an algorithm grounded in ICD-9 codes that classifies 12 preoperative comorbidities. This index assigns scores ranging from 0 to 24, with higher scores being indicative of a more severe comorbidity status. Preoperative neurological status was dichotomized using the ASIA Impairment Scale, categorized as “neurological deficits” with grades A, B, C, D, or “no neurological deficits” with grade E, encompassing individuals with prior, but not current, deficits.

We defined previous systemic therapy as all forms of nonsurgical adjuvants or non-radiotherapeutic adjuvants administered before the surgery, including cytotoxic, immunologic, metabolic, or hormonal therapy. Surgical procedures were categorized into four groups as follows: (1) vertebrectomy or corpectomy with stabilization, (2) decompression and stabilization, (3) decompression alone, or (4) stabilization alone. The Katagiri grouping was based on histology, with slow growth encompassing hormone-dependent breast cancer, hormone-dependent prostate cancer, malignant lymphoma, malignant myeloma, and thyroid cancer. Moderate growth included non-small cell lung cancer with molecularly targeted therapy, hormone-independent breast cancer, hormone-independent prostate cancer, renal cell carcinoma, sarcoma, other gynecological cancer, and others. Rapid growth comprised other lung cancer, colon and rectal cancer, gastric cancer, hepatocellular carcinoma, pancreatic cancer, head and neck cancer, other urological cancer, esophageal cancer, malignant melanoma, gallbladder cancer, cervical cancer, and cases of unknown origin.

### 2.3. CT Body Composition Assessment

A CT scan of the abdominal region was performed, utilizing cutting-edge 16-slice or 64-slice CT scanners manufactured by Siemens (Erlangen, Germany) or GE Healthcare (Chicago, IL, USA). The imaging protocol incorporated precise parameters, including a 5 mm slice thickness, 15 mm/s table feed, 1.5 pitch, 120 kVP tube voltage, variable tube current (with a maximum of 450) mAs, and sagittal and coronal reconstructions at a 2 mm thickness with 2 mm intervals. These scanners underwent rigorous annual testing in accordance with the stringent standards set forth by the American Association of Physicists in Medicine (AAPM) and the American College of Radiology (ACR). Robust clinical quality assurance measures were implemented to ensure the uniformity and reproducibility of the scans.

Measurements were performed at the mid-portion of the L4 vertebral body level using an in-house automated machine learning algorithm designed for body composition analysis. This artificial intelligence algorithm segmented and quantified the following body composition parameters: the cross-sectional area (CSA) of subcutaneous adipose tissue (SAT) (Figure 1), visceral adipose tissue (VAT) (Figure 2), and paraspinal/abdominal muscle (Figure 3). Figure 4 shows the full segmentation of an axial image. Sarcopenia was defined by the total muscle CSA (cm^2^) divided by height squared (m^2^) with cutoff values of <52.4 cm^2^/m^2^ for men and <38.5 cm^2^/m^2^ for women [40]. The process and machine learning-produced data were overseen by a singular, highly trained imaging analyst (CGB), maintaining strict blindness to clinical details and patient outcomes. Segmentation adjustments were executed using the Horos DICOM viewer (version 6.5.2, www.horosproject.com, 1 June 2021), under the supervision of two senior fellowship-trained musculoskeletal radiologists (MT, MAB).

### 2.4. Outcomes

The outcome variables were (1) LOS (days), (2) postoperative complications within 30 days, and (3) reoperations until final follow-up or death. We considered the following postoperative complications within 30 days: venous thromboembolism, pneumonia, myocardial infarction, urinary tract infection, sepsis, wound infection and/or dehiscence [16,38,41,42]. Only symptomatic venous thromboembolisms were taken into account, manifesting as swelling, redness, pain in the lower extremities, or respiratory issues. This was defined as any symptomatic pulmonary embolism or symptomatic distal or proximal deep vein thrombosis diagnosed within 30 days of the surgery, using the following diagnostic procedures: venography, impedance plethysmography, pulmonary arteriography, chest CT, ventilation-perfusion lung scan, and vascular ultrasound [38]. Reoperation was defined as unplanned surgical reintervention to the initial surgical site. Death was outlined as mortality resulting from any cause, as we expected that the majority of deaths were related to metastatic disease in these terminal patients. The date of death was determined using the Social Security Death Index and by reviewing medical records. There was no loss to follow up within 30 days, and the median follow-up was 9 months (interquartile range [IQR], 3–25 months). Follow-up was verified until 15 May 2020.

### 2.5. Statistical Analysis

Continuous variables are presented as medians with IQRs and categorical variables as frequencies with percentages. Linear regression was used to test continuous outcomes (LOS) and logistic regression for categorical outcomes (complications within 30 days and reoperations). Each separate body composition measure with *p* < 0.10 was included in multivariate analyses while controlling for all clinical variables that were *p* < 0.10 in bivariate analysis. Collinearity was tested, and BMI was excluded from the multivariate analyses because of high collinearity with the body composition measures. Multiple imputation (*n* = 40) was applied for the following missing variables: BMI in 11 patients (6%) and ASIA score in 2 patients (1%). No multiple imputation was applied for the body composition measures as this was the variable of interest. A two-tailed *p*-value of <0.05 was considered significant. All statistical analyses were performed using R version 3.6.3 (The R Foundation, Vienna, Austria), R studio version 1.3.887 (RStudio, Boston, MA, USA), and Stata 15.0 (StataCorp LP, College Station, TX, USA). Mendeley Desktop Version 1.19.4 (Mendeley Ltd., London, UK) was used as a reference management software.

## 3. Results

### 3.1. Patients and Characteristics

Of the included 196 patients, 123 (63%) were males and 73 (37%) were females with a median age of 62 years (IQR, 53–70) and a BMI of 26 kg/m^2^ (IQR, 23–30). Forty-two percent of the patients had sarcopenia. The median preoperative albumin value was 3.8 g/dL (IQR, 3.4–4.2). The median duration of the primary diagnosis until metastatic operation was 397 days (IQR, 26–1464). Additional comorbidities were present in 127 (65%) patients. Additional metastases besides the treated spinal lesion were found in 138 (70%) of patients. The most common primary tumors included renal cell (14%), lung (13%), breast (7.7%), and multiple myeloma (7.1%). When categorized according to the primary tumor growth, rapid tumors were most frequent (41%; 81/196), followed by moderate tumor growth (34%; 67/196) and slow moderate growth (25%; 48/196). Most patients had spinal pain (88%; 172/196). The preoperative ASIA impairment scale was almost equal between patients presenting neurological deficits (45%; 88/196) and patients without neurological deficits (55%; 108/196). The most frequent metastases region was thoracic spine (54%; 105/196), followed by the lumbar region (28%; 54/196), cervical region (14%; 28/196), and combined (4.6%; 9/196). Over half of the patients received previous systemic therapy (55%; 107/196) and had a completed pathological fracture (54%; 106/196). Most patients underwent one spinal level undergoing surgery (47%; 93/196), followed by three or more levels (36%; 71/196) and two levels (16%; 32/196). The most common surgical approach was posterior (86%; 169/196) and anterior (11%; 22/196), while a combined approach was only performed in five patients (2.6%). Both complete vertebrectomy or corpectomy with stabilization (40%; 78/196) and partial vertebrectomy or corpectomy with stabilization were performed (39%; 76/196) and were the most often performed surgeries, followed by decompression (14%; 28/196) and stabilization (7.1%; 14/196) (Table 1).

### 3.2. Length of Stay (LOS)

The median length of stay was 9 days (IQR, 6–13). During bivariate analysis, four clinical variables were associated with increased LOS: lower albumin level, additional comorbidity, three or more spine level undergoing operation, and a combined surgical approach (Appendix A). No body composition measurements were associated with the length of stay (all *p* > 0.10).

### 3.3. Postoperative Complications within 30 Days

Postoperative complications within 30 days occurred in 61 (31%) patients. During bivariate analysis, lower muscle CSA (OR [95% CI] = 0.99 [0.98–0.99], *p* = 0.03) was associated with increased postoperative complications within 30 days, and there was a trend for lower SAT CSA (OR [95% CI] = 0.99 [0.99–1.01], *p* = 0.06). The following four clinical variables were associated with increased postoperative complications: lower BMI, lower albumin, ASIA score, and thoracic metastases. BMI was not included in multivariate analysis due to high collinearity with the body composition measurements. With multivariate analysis, lower muscle CSA (OR [95% CI] = 0.99 [0.98–0.99], *p* = 0.047) was associated with increased postoperative complication rate after controlling for albumin, ASIA score, previous systemic therapy, and thoracic metastases (Table 2). SAT CSA lost significance when analyzing 30-day postoperative complications after controlling for the four clinical variables (*p* = 0.06).

### 3.4. Reoperations

Reoperations occurred in 31 (16%) patients. No body composition measurements were associated with reoperation (all *p* > 0.10).

## 4. Discussion

Our study showed that low muscle CSA obtained from opportunistic abdominal CTs using automated machine learning algorithms can predict post-surgical complications in patients undergoing spine surgery for metastatic disease, independent of other established risk factors. The opportunistic assessment of body composition using artificial intelligence could therefore become an imaging biomarker for risk assessment in cancer patients with metastatic disease who routinely undergo abdominal CTs for staging and surveillance.

Body composition assessments using opportunistic CTs have been increasingly used for prognostication in patients with malignancies who are undergoing surgery [24,28,31,32]. However, only few studies have assessed the impact of body composition measures on postoperative adverse events, which are critical for the shared decision making between surgeons and potential surgical candidates [24]. To our knowledge, this study is the first to determine the predictive value of body composition assessed using machine learning technology on opportunistic CTs to determine LOS, postoperative complications within 30 days of surgery, and reoperations in patients with spinal metastases undergoing surgical treatment. We demonstrated that lower muscle CSA was associated with increased postoperative complications within 30 days, suggesting that body composition can be used in conjunction with clinical factors to better predict adverse outcomes after surgery.

The association between lower muscle CSA and greater postoperative complications within 30 days is consistent with other oncology studies [43]. In a review of five studies that assessed complications in patients with abdominal and genitourinary malignancies, low muscle CSA was an important prognostic factor for complications and survival [43]. Our study supports these findings by including the largest spinal cohort to date with various primary malignancies, controlling for multiple clinical cofounders, and establishing a convenient and reliable method of extracting body composition measures via the use of an in-house automated algorithm.

Cachexia, a systemic tissue-wasting process that affects the quality and amount of muscle tissue, may be the cause of body composition alterations found in individuals with metastatic illness [44]. Cancer causes a hypermetabolic state caused by a mix of tumor metabolism and systemic inflammation which alters the homeostasis of the body [45]. This combined with cancer-related fatigue, anorexia, and limited functional status leads to a depletion of skeletal muscle [43,46]. This may be mediated by the inflammatory cytokines such as tumor necrosis factor-alpha or IL-6, which exert catabolic effects on muscle by stimulating protein loss in muscle cells [47,48]. The catabolic effect can lead to greater skeletal muscle loss than in other tissues. A better understanding of the role of these inflammatory cytokines and the molecular mechanism behind disproportionate skeletal muscle loss in elderly, surgical, and oncologic populations may help us to understand the relation of this finding to the outcome. Additionally, this understanding may also point us towards other body composition parameters that can further help us improve prognostic tools, treatment modalities, and preventative measures.

### 4.1. Future Implications

The availability of preoperative CT scans in cancer patients and automated machine learning algorithms that can quickly and reliably extract body composition measures make opportunistic body composition analysis an attractive imaging biomarker for the prediction of outcomes. However, future prospective studies should determine the value of these CT data in the clinical setting with standardized CT protocols in large data sets from multiple diverse centers. A prognostication tool that considers both survival and adverse outcomes and is proven to be beneficial in the clinical setting could then be added to the existing electronic healthcare software, automatically inputting the clinical and algorithm obtained body composition data into the model. By assessing the likelihood of survival and different postoperative complications, it will allow surgeons and patients to make informed decisions regarding the best approach to the treatment of their metastatic disease.

### 4.2. Limitations

This study has several limitations. This was a retrospective study from one tertiary medical center, causing the inevitable risk of selection and confounding bias; in addition, surgery is typically not the initial treatment for spine metastases. Instead, it is utilized when there are neurological complications. Spinal cord compression and impending or unstable pathological fractures are often urgent indications for surgery, and spending time to predict postoperative outcomes may not be useful or appropriate. However, creating the best tools possible to aid physicians and patients in the shared decision-making process will be useful to prevent the postoperative morbidity and mortality associated with surgery. Additionally, improving or maintaining the quality of life is recognized as an important outcome when evaluating a patient for the surgical management of spinal metastases [7]. Our hospital did not assess the quality of life during routine case visits, which could have been a valuable inclusion in this study, especially given the frailty of the patient population. Furthermore, this study does not stratify the findings between male and female patients. It has been shown that there are inherent intrinsic differences in the muscle and body composition measures of males and females [49,50]. The cohort in this study was 63% male, and is unlikely to affect the utility of this technology for outcome prediction. Future prospective studies using opportunistic CTs for body composition measurements across various populations with sex stratification would be required to assess the generalizability of these findings. Strengths of our study include the large longitudinal cohort with detailed clinical and outcome measures, and the detailed assessment of body composition.

## Figures and Tables

**Figure 1 diagnostics-14-00844-f001:**
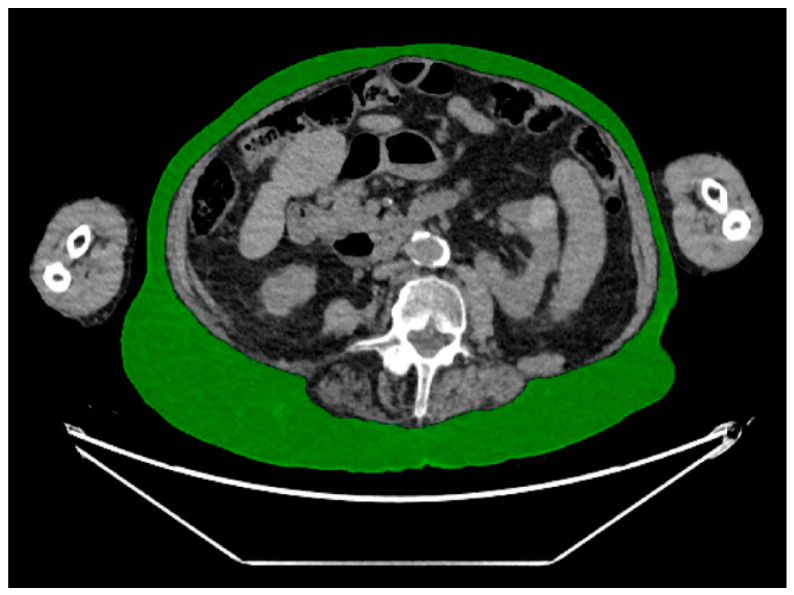
Representative axial CT image of the abdomen, with green representing subcutaneous adipose tissue.

**Figure 2 diagnostics-14-00844-f002:**
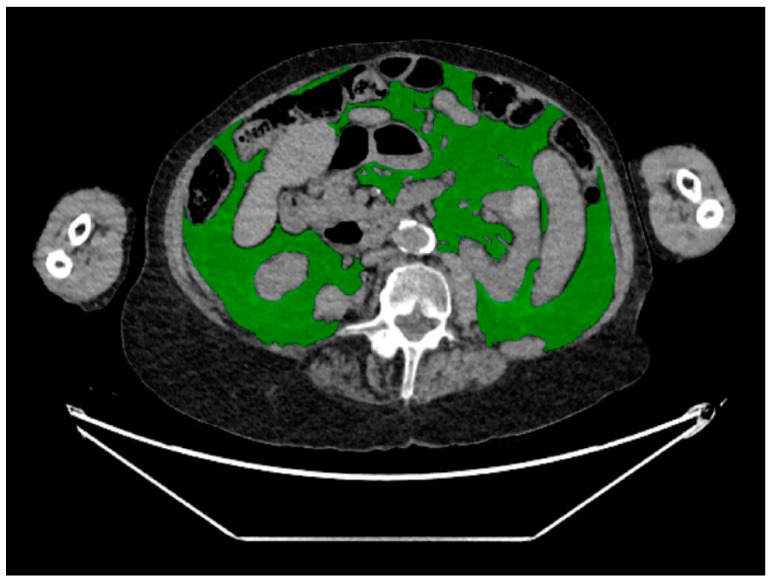
Representative axial CT image of the abdomen, with green representing visceral adipose tissue.

**Figure 3 diagnostics-14-00844-f003:**
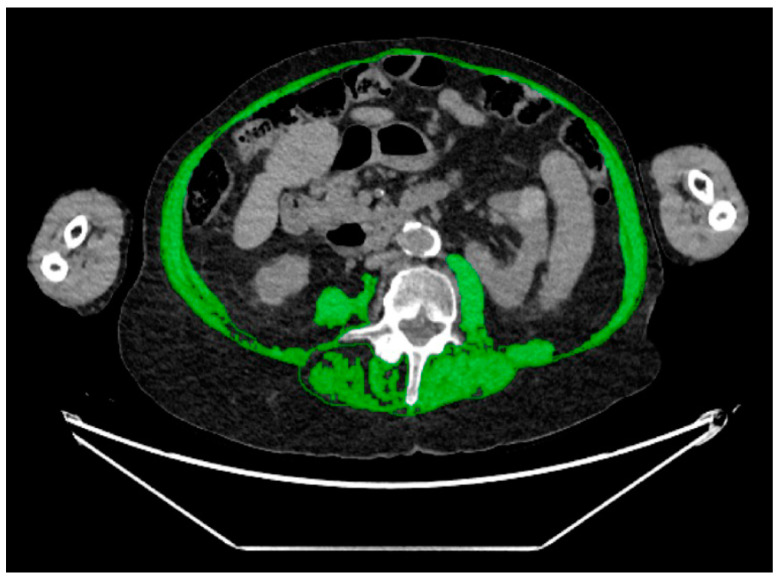
Representative axial CT image of the abdomen, with green representing muscle mass.

**Figure 4 diagnostics-14-00844-f004:**
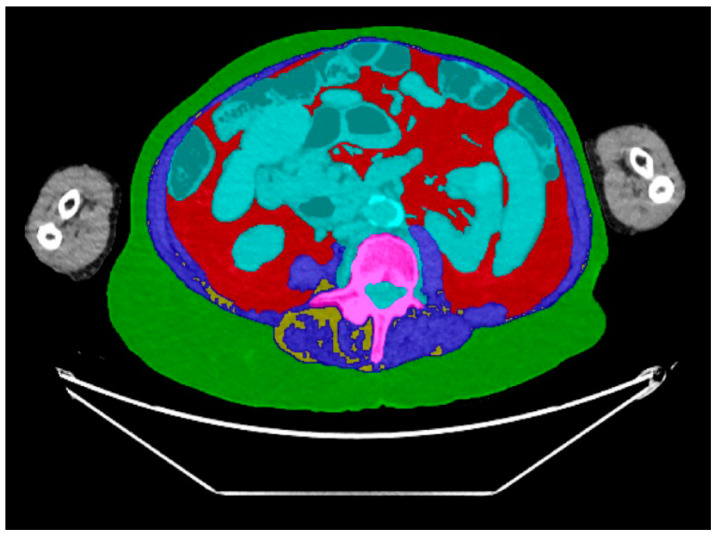
Axial CT images of the abdomen with the segmentation of body composition parameters using artificial intelligence. Green: subcutaneous adipose tissue, red: visceral adipose tissue, blue: muscle mass, turquoise: bowel and blood vessels, pink: vertebral body, yellow: intermuscular adipose tissue.

**Table 1 diagnostics-14-00844-t001:** Baseline characteristics of patients surgically treated for spinal metastases (*n* = 196).

Variables	Spine (*n* = 196)
	** *Median (IQR)* **
Age (years)	62 (53–70)
Body mass index (in kg/m^2^) ^a^	26 (23–30)
Duration primary diagnosis untill metastatic operation (days)	397 (26–1464)
Albumin (g/dL)	3.8 (3.4–4.2)
	** *% (n)* **
Male	63 (123)
Additional Modified Charlson comorbidity ^b^	65 (127)
Primary tumor growth ^c^	
Slow	25 (48)
Moderate	34 (67)
Rapid	41 (81)
Additional metastases ^d^	70 (138)
Spinal pain	88 (172)
ASIA impairment scale (preoperative) ^a^	
Neurological deficit (A, B, C, or D)	45 (88)
No neurological deficit (E)	55 (108)
Metastases region	
Thoracic	54 (105)
Lumbar	28 (54)
Cervical	14 (28)
Combined	4.6 (9)
Previous local radiotherapy	33 (64)
Previous systemic therapy	55 (107)
Pathological fracture	54 (106)
Number of spine levels undergoing operation	
1	47 (93)
2	16 (32)
3 or more	36 (71)
Type of surgery	
Vertebrectomy or corpectomy with stabilization	40 (78)
Decompression and stabilization	39 (76)
Decompression	14 (28)
Stabilization	7.1 (14)
Surgical approach	
Posterior	86 (169)
Anterior	11 (22)
Combined	2.6 (5)
Two-staged procedure	1.0 (2)
** *Body composition measures ^a^* **	** *Median (IQR) or % (n)* **
Subcutanous adipose tissue	
Area (cm^2^)	249 (180–320)
Visceral adipose tissue	
Area (cm^2^)	124 (75–211)
Muscle	
Area (cm^2^)	140 (116–165)
** *Outcomes* **	
Length of stay in days	9 (6–13)
Postoperative complications within 30 days	31% (61)
Reoperations	16% (31)

IQR = interquartile range; kg/m^2^ = kilogram per square meter; g/dL = gram per deciliter; ASIA = American Spinal Injury Association; cm^2^ = square centimeters. ^a^ Body mass index was available in 94% patients (185), ASIA impairment scale in 99% patients (194), SAT area in 80% patients (157), VAT area in 100% of the patients (196), and muscle cross sectional area in 80% patients (157). ^b^ These values were based on any additional comorbidity on top of the metastatic disease score according to the modified Charlson Comorbidity Index. ^c^ Based on histology groupings; slow growth includes hormone-dependent breast cancer, hormone-dependent prostate cancer malignant lymphoma malignant myeloma, and thyroid cancer; moderate growth includes non-small cell lung cancer with molecularly targeted therapy, hormone-independent breast cancer, hormone-independent prostate cancer, renal cell carcinoma, sarcoma, other gynecological cancer, and others; and rapid growth includes other lung cancer, colon and rectal cancer, gastric cancer, hepatocellular carcinoma, pancreatic cancer, head and neck cancer, other urological cancer, esophageal cancer, malignant melanoma, gallbladder cancer, cervical cancer, and unknown origin. ^d^ Any metastasis outside of the lesion were treated.

**Table 2 diagnostics-14-00844-t002:** Multivariable logistic regression analysis of the muscle area for 30-day postoperative complications after surgery for spinal metastases using pooled imputed data.

*Variables*	*Odds Ratio (95% CI)*	*Standard-Error*	*p-Value*
Albumin	0.42 (0.21; 0.82)	0.143	**0.01**
ASIA impairment scale (preoperative)			
Neurological deficit (A, B, C, or D)	*Reference value*		
No neurological deficit (E)	0.65 (0.30; 1.41)	0.258	0.28
Metastases region			
Thoracic	*Reference value*		
Lumbar	0.88 (0.39; 1.98)	0.363	0.76
Cervical	0.12 (0.02; 0.55)	0.093	**0.01**
Combined	0.11 (0.01; 1.17)	0.133	0.07
Previous systemic therapy	1.27 (0.58; 2.78)	0.508	0.55
Muscle area (cm^2^)	0.99 (0.98; 0.99)	0.006	**0.047**

*CI = confidence interval*. ***Bold** p-values are <0.05*.

## Data Availability

The datasets presented in this article are not readily available because they are part of an ongoing study.

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
