# Peer review of "Opportunistic CT for Prediction of Adverse Postoperative Events in Patients with Spinal Metastases"

_diagnostics, 2024, doi:10.3390/diagnostics14080844_

Round 1

Reviewer 1 Report

Comments and Suggestions for Authors

Dear Editor,

-          Although the article is interesting, it should be considered that reduced skeletal muscle mass is a key feature of cancer-associated cachexia, and CT scanning is an excellent tool for evaluating axial skeletal muscle mass. Also, the decrease in muscle mass affects the outcome of cancer patients (decrease in treatment effectiveness, increase in complications after cancer surgery, and mortality). Therefore, if the majority of patients in this study had cachexia, more complications would have been expected. In my opinion, the authors should explain more about this, why did they decide to evaluate it when it is clear that there are more complications in patients with cachexia? Was their intention to evaluate it as an indicator of cachexia in these patients at the end stage? What was the relationship between muscle mass reduction and BMI and weight of patients?

-          Also, in lines 214 to 216, the following sentence should be corrected. ‘The preoperative ASIA impairment scale was almost equal between patients presenting with neurological deficits (45%; 88/196) and patients without neurological deficits (55%; 106/196). The total number of patients is 194.

Yours sincerely,

Author Response

We would like to thank the Reviewers for their time to review our manuscript and for their helpful comments. Please find the detailed responses below and the corresponding revisions/corrections in red in the re-submitted files. We believe this work adds valuable information to the growing body of literature in the field of artificial intelligence to promote patient care.

Comment 1:

Although the article is interesting, it should be considered that reduced skeletal muscle mass is a key feature of cancer-associated cachexia, and CT scanning is an excellent tool for evaluating axial skeletal muscle mass. Also, the decrease in muscle mass affects the outcome of cancer patients (decrease in treatment effectiveness, increase in complications after cancer surgery, and mortality). Therefore, if the majority of patients in this study had cachexia, more complications would have been expected. In my opinion, the authors should explain more about this, why did they decide to evaluate it when it is clear that there are more complications in patients with cachexia? Was their intention to evaluate it as an indicator of cachexia in these patients at the end stage? What was the relationship between muscle mass reduction and BMI and weight of patients?

Response 1:

We appreciate the Reviewer’s comments on cancer-associated cachexia and the associated complications. There are no body composition measures to define cachexia, however, there are published threshold to determine sarcopenia. Based to the Reviewer’s suggestion, we evaluated for the presence of sarcopenia in our cohort by total muscle CSA (cm2) divided by height squared (m2) with cutoff values of <52.4 cm2/m2 for men and <38.5 cm2/m2 for women, and found that sarcopenia was present in only 42% of our patients. We have added this information in the Results section. (Page 4, lines 165-166; Page 5, lines 210-211) As less than half of our patients had sarcopenia, our results of a positive association of low muscle mass and postoperative complications even in patients without sarcopenia is even more relevant. BMI was not included in our multivariate analyses due to high collinearity with body composition measures. (Page 5, lines 197-198)

Comment 2:

Also, in lines 214 to 216, the following sentence should be corrected. ‘The preoperative ASIA impairment scale was almost equal between patients presenting with neurological deficits (45%; 88/196) and patients without neurological deficits (55%; 106/196). The total number of patients is 194.

Response 2:

Thank you for recognizing this oversight. We have made the correction to appropriately reflect the correct number and included it in the new Table 1.

Reviewer 2 Report

Comments and Suggestions for Authors

The article aimed to evaluated the value of body composition measures to predict hospital lenght of stay in patients with spinal metastases that could be of interest to a large number of professional. The following suggestion may improve the quality of your work. 

Introduction

-please add more references that support your work

- the primary and secondary aim should be describe and the relative primary and secondary outocomes measures

method

- the nature of the study should be explain, observational?

- who performed the enrollment and the evaluation?

Result

- a table with the charatteristics of the sample should be added 

Discussion

- a gender stratification should be added, or this should be added in the limitation section. Infact there are instrinsic sex-base differences in the muscle morphology and charatteristics. Please take in to consideration thse two articles in your limitaion section

Deodato, M., Saponaro, S., Šimunič, B. et al. Sex-based comparison of trunk flexors and extensors functional and contractile characteristics in young gymnasts. Sport Sci Health 20, 147–155 (2024). https://doi.org/10.1007/s11332-023-01083-7

Lohr C, Schmidt T, Braumann KM, Reer R, Medina-Porqueres I. Sex-Based Differences in Tensiomyography as Assessed in the Lower Erector Spinae of Healthy Participants: An Observational Study. Sports Health. 2020 Jul/Aug;12(4):341-346. doi: 10.1177/1941738120917932. Epub 2020 Jun 8. PMID: 32511080; PMCID: PMC7787569.

Author Response

We would like to thank the Reviewers for their time to review our manuscript and for their helpful comments. Please find the detailed responses below and the corresponding revisions/corrections in red in the re-submitted files. We believe this work adds valuable information to the growing body of literature in the field of artificial intelligence to promote patient care.

Comment 1:

Introduction

-please add more references that support your work

- the primary and secondary aim should be described and the relative primary and secondary outcomes measures

Response 1:

We thank the Reviewer for this comment and have added more references as requested. The aims and outcome measures have also been added to the Introduction. (Page 2, lines 80-83)

Comment 2:

method

- the nature of the study should be explained, observational?

- who performed the enrollment and the evaluation?

Response 2:

We appreciate this feedback. The study is retrospective and observational and this has been added to the Methods section. (Page 3, line 88)

All patients who had undergone surgery for spinal metastases were included in the study. Subsequently, two authors (NDK, PKT) assessed the 364 patients that underwent surgery, and 168 of these patients were excluded for not have an evaluable CT scan. CT scans were analyzed using our AI algorithm. This information has been added to the Methods section. (Page 3, lines 100-103)

Comment 3:

Result

- a table with the characteristics of the sample should be added 

Response 3:

We thank the Reviewer for this comment and added a new Table 1 with this information.

Comment 4:

Discussion

- a gender stratification should be added, or this should be added in the limitation section. In fact there are intrinsic sex-base differences in the muscle morphology and characteristics. Please take into consideration these two articles in your limitation section

Deodato, M., Saponaro, S., Šimunič, B. et al. Sex-based comparison of trunk flexors and extensors functional and contractile characteristics in young gymnasts. Sport Sci Health 20, 147–155 (2024). https://doi.org/10.1007/s11332-023-01083-7

Lohr C, Schmidt T, Braumann KM, Reer R, Medina-Porqueres I. Sex-Based Differences in Tensiomyography as Assessed in the Lower Erector Spinae of Healthy Participants: An Observational Study. Sports Health. 2020 Jul/Aug;12(4):341-346. doi: 10.1177/1941738120917932. Epub 2020 Jun 8. PMID: 32511080; PMCID: PMC7787569.

Response 4:

We thank the Reviewer for this important point and the suggested references. We have included sex differences in muscle in our limitation section and included the suggested references. (Page 8 lines 334-340)

Round 2

Reviewer 1 Report

Comments and Suggestions for Authors

Dear Editor,

It is suitable for publication in its current form.

Yours Sincerely,